 

# Using high-resolution global climate models from the PRIMAVERA project to create a European winter windstorm event set

Julia F. Lockwood[1], Galina S. Guentchev[1], Alexander Alabaster[2], Simon J. Brown[1], Erika J. Palin[1], Malcolm J. Roberts[1], Hazel E. Thornton[1]

[1]Met Office Hadley Centre, Exeter, EX1 3PB, UK
[2]AON, London, EC3V 4AN, UK

*Correspondence to*: Julia F. Lockwood (julia.lockwood@metoffice.gov.uk)

**Abstract.** PRIMAVERA was a European Union Horizon 2020 project whose primary aim was to generate advanced and well-evaluated high-resolution global climate model datasets, for the benefit of governments, business and society in general.
Following consultation with members of the insurance industry, we have used a PRIMAVERA multi-model ensemble to generate a European winter windstorm event set for use in insurance risk analysis, containing approximately 1300 years of windstorm data.

To create the storm footprints for the event set, the storms in the PRIMAVERA models are identified through tracking. A
method is developed to separate the winds from storms occurring in the domain at the same time. The wind footprints are bias corrected and converted to 3-s gusts onto a uniform grid using quantile mapping. The distribution of the number of model storms per season as a function of estimated loss is consistent with re-analysis, as are the total losses per season, and the additional event set data greatly reduces uncertainty on return period magnitudes. The event set also reproduces the temporally clustered nature of European windstorms.

Since the event set is generated from global climate models, it can help to quantify the non-linear relationship between large-scale climate indices such as the North Atlantic Oscillation (NAO) and windstorm damage. Although we find only a moderate positive correlation between extended winter NAO and storm damage in northern European countries (consistent with re-analysis), there is a large change in risk of extreme seasons between negative and positive NAO states. The intensities of the
most severe storms in the event set are, however, sensitive to the gust conversion/bias correction method used, so care should be taken when interpreting the expected damages for very long return periods.



## 1 Introduction

Winter European windstorms are the costliest natural hazard over Europe in terms of insured losses, capable of inflicting billions of dollars of loss per event. According to Swiss Re (2018), of *all* hazards affecting Europe since 1970, the top 5 highest loss events (after adjusting for inflation to 2017 USD) were all winter windstorms: Daria (1990, 8.7bn USD), Lothar (1999, 8.5bn USD), Kyrill (2007, 7.2bn USD), 87J (1987, 6.7bn USD) and Vivian (1990, 6.5bn USD).

Insurance and re-insurance companies must estimate the financial risk posed by these events to ensure they are able to pay out the resulting claims. A common technique used to analyse this risk is catastrophe modelling, where the insured losses due to a particular hazard are estimated by combining the hazard footprint with the clients' exposure and policy data. For European windstorms, the footprint is defined as the maximum wind gust associated with the storm over a 72 hour period (where the gust is defined as the maximum 3-s average wind speed at 10m height, according to World Meteorological Organization observing practices; WMO, 2018). The catastrophe models can be run using either a single event footprint (e.g. a notable past event or plausible extreme future event), which can be useful for verifying the catastrophe model or understanding the vulnerability of the insurer, or an event set – a set of thousands of event footprints – to estimate large return period losses. The 1 in 200 year loss is of particular interest since European law requires that EU based insurers hold enough capital to withstand losses at this level (Solvency II, 2009). Since observational and re-analysis datasets typically span decades rather than centuries, event set footprints must be constructed using either statistical models, dynamical (climate) models, or a combination of the two.

In purely statistical methods, footprints can be generated in a variety of ways: the geospatial dependencies of observed extreme gusts can be captured in statistical models, allowing footprints to be generated from a random seed value (e.g. Youngman and Stephenson, 2016); or historical footprints can be 'perturbed' to generate a number of new events that differ slightly from past ones, so are deemed to be physically plausible (e.g. Welker at al., 2020). Alternatively, a set of storm *tracks* can be generated based on the properties of historical tracks, and new footprints generated from the statistical relationship between tracks and footprints (e.g. Sharkey et al., 2019; Sharkey et al., 2020).

Dynamical methods using climate models are commonly used in estimations of windstorm risk, in particular with regards to estimating the effect of climate change (e.g., Leckebusch et al., 2007; Della-Marta and Pinto, 2009), but the coarse resolution of the models means that to generate an event set for use in an industry catastrophe model, statistical downscaling and bias correction of the footprints are often required. Examples include Haylock (2011), who extracted their windstorm event set from an ensemble of regional climate models at 25km resolution, driven by coarse resolution global climate models. The resulting footprints were downscaled further to 7km by accounting for changes in roughness and orography between the 25km and 7km grids. Finally, the climate model footprints were bias corrected by applying quantile mapping to historical footprints





on the same 7km grid. The Windstorm Information Service (WISC) project (Steptoe, 2017), generated an event set from the UPSCALE atmosphere-only global climate models at approximately 25 km resolution (Mizielinski et al., 2014), which were then downscaled and bias corrected to 4.4 km, again using quantile mapping to historical storms on the target grid (but without correcting for roughness and orography). Dynamical event sets can also be produced from medium to long range ensemble prediction systems (e.g. Osinski et al., 2016; Walz and Leckebusch, 2019), where the chaotic nature of the atmosphere means

that after a few days to weeks individual storms in each ensemble member will be largely independent.

Advantages of the statistical methods include low computational costs, but they are ultimately model based on a short observational time period (typically less than 50 years for re-analysis and observational datasets), and although they can be used to generate thousands of footprints, the frequency of each of those footprints is difficult to estimate. The dynamical-

statistical methods are computationally expensive, but the frequency of the windstorms can easily be taken directly from the model, and they are likely to be physically plausible. However, it is known that low resolution global climate models suffer from biases in the North Atlantic storm track, where it is too zonal or displaced southwards (Zappa et al., 2013), meaning that event sets with low resolution driving models could lead to errors in the spatial distribution of estimated storm loss.

In this paper, we describe an event set produced from the PRIMAVERA high-resolution global climate model ensemble. PRIMAVERA (Process-based climate simulation: Advances in high-resolution modelling and European climate risk assessments; https://www.primavera-h2020.eu/) was a European Union Horizon 2020 project whose aim was to generate advanced and well-evaluated high-resolution global climate model datasets, and to interpret or process this data to meet the needs of sectors such as energy, water management, agriculture, transport, health, and finance/insurance.


The event set is generated from the historical atmosphere-only experiments from five different models, at both a standard CMIP6-type resolution (typically 100 km) and at a significantly higher resolution (towards 25 km), producing approximately 1300 years of model data. Climate models run at these higher resolutions suffer less from the North Atlantic storm track biases found in models at typical CMIP5-generation and earlier resolutions (~200–300km) (e.g. Zappa et al., 2013; Baker et al., 2019;

Priestley et al., 2020), which should result in more realistic storm frequencies, spatial distributions and intensities. Possible reasons for this bias reduction include improvements in the representation of orographic drag which improves the simulation of climatological stationary Rossby waves (Pithan et al., 2016); increased latent heating leading to a more realistic intensification of extra-tropical cyclones (Willison et al., 2013) and a more tilted storm track (Tamarin-Brodsky and Kaspi, 2017); improvements in European blocking which helps steer the storm track northwards rather than into central Europe

(Schiemann et al., 2020); and sharpening of sea surface temperature (SST) gradients and reductions in SST biases leading to changes in low level baroclinicity and latent heat supply (Small et al., 2019).



Since the PRIMAVERA models are global, it is also possible to associate each storm to large scale climate indices such as the North Atlantic Oscillation (NAO). Given recent advances in NAO prediction on seasonal (Scaife et al., 2014) to annual
(Dunstone et al., 2016) and multi-annual (Athanasiadis et al., 2020; Smith et al., 2020) timescales, this gives insurers the possibility of using their catastrophe models in a predictive mode, assessing the change in storm risk for a given NAO forecast.

The aim of this paper is to describe the method used to create the event set from PRIMAVERA models and to show how it compares to re-analysis. The method involves first identifying the storms using a tracking algorithm, then extracting the model
surface winds associated with each storm to make the footprint. The footprints from different climate models are then re-gridded to a common 0.25°×0.25° grid, and the model winds are bias corrected and converted to gusts using quantile mapping. The model and re-analysis data used are described in Sect. 2, and Sect. 3 gives a full description of the method. In Sect. 4 we show the comparison with re-analysis, and also the relationship between storm loss and the NAO. Section 5 discusses the sensitivity of footprint intensity to the bias correction method, and conclusions are given in Sect. 6.

**2 Data**

**2.1 PRIMAVERA model data**

The event set is made from the highresSST-present PRIMAVERA experiments. These simulations are atmosphere only, covering the period 1950–2014, and use the historical forcings detailed in the HighresMIP protocol (Haarsma et al., 2016, Table 1). The lower boundary was forced by the daily, ¼° Hadley Centre Global Sea Ice and Sea Surface Temperature
(HadISST.2.2.0; Kennedy et al., 2017) dataset, with area-weighted regridding used to map this to each model grid.

The PRIMAVERA models used for the event set are summarised in Table 1. Each model was run at both a standard CMIP6-type resolution (typically 100 km) and at a significantly higher resolution (towards 25 km), and some models ran multiple ensemble members. Note that although the CMIP6-type resolution is often referred to as 'low-resolution', these resolutions
were considered relatively high for global models of the CMIP5 generation. For example, the CMIP5 model HadGEM2-ES on the N96 grid (~135km grid spacing at mid-latitudes) was categorised by Zappa et al. (2013) as one of the higher resolution CMIP5 models with a small bias in the North Atlantic storm track.

A windstorm footprint is defined as the maximum *3-s gust* associated with the storm over a 72-hour period, but because only
two PRIMAVERA models outputted maximum gusts we instead extract daily maximum surface (10m) winds (sfcWindmax) from the PRIMAVERA models, and convert from winds to gusts as described in Sect. 3.1.3. Only data for October–March is extracted, to cover the extended winter season. These winter storms tend to be associated with extra-tropical cyclones and span a larger spatial area compared to smaller, convective storms that occur in summer. 6-hourly 850hPa u and v winds were extracted to calculate the relative vorticity needed for the tracking algorithm to identify storms, described in Sect. 3.1.1.





## 2.2 Re-analysis data

To validate the PRIMAVERA event set, footprints were also made from the ECMWF Reanalysis 5th Generation (ERA5; Copernicus Climate Change Service, 2017) wind gusts. This data set covers the period 1979–2014, on a 0.25°×0.25° grid (~18km grid spacing at 50° N). Hourly maximum gusts for October–March were extracted and converted to daily maxima for fair comparison with PRIMAVERA models.

Tracking to identify the re-analysis storms (Sect. 3.1.1) was performed on an earlier version of ECMWF Re-analysis, ERA-Interim (Dee et al., 2011) since ERA5 tracks were unavailable at the time. Since northern hemisphere cyclone tracks have been shown to match well between re-analyses, particularly for intense cyclones (Hodges et al., 2011), the inconsistency between the tracks and gust data is expected to be small.

## 3 Methods

### 3.1 Generating the footprints

To comply with industry standards, a windstorm footprint is defined as the maximum 3-s gust associated with the storm over a 72-hour period. The footprint domain is defined as 25° W to 40.5° E in longitude, and 34.4° N to 71.5° N in latitude (Fig. 1).

Storms are first identified with the tracking algorithm (Sect. 3.1.1). One 72-hour footprint is produced per track, despite typical track lengths being longer than 72 hours. Following Roberts et al. (2014), for each track, the central day of the 72-hour period over which to take the maximum gusts is identified by finding the day of the maximum 10m wind speed over land within 3° of the track centre (output by the tracking algorithm).

Often two or more tracks will have the same or overlapping 72-hour periods identified for their footprints. Taking the maximum winds/gusts over the whole domain for the specified 72-hour period for each event would result in several cyclones being present in a single footprint, and many cyclones would be double counted in the resulting event set. The footprints are therefore separated as described in Sect. 3.1.2. Finally, the footprints must be re-gridded to a common, high-resolution grid, converted from maximum winds to maximum 3-s gusts, and bias corrected; this is described in Sect. 3.1.3.





| Institution | MOHC, UREAD, NERC | EC-Earth KNMI, SHMI, BSC, CNR | CERFACS | MPI-M | CMCC |
|---|---|---|---|---|---|
| Model name | HadGEM3-GC3.1 | EC-Earth3P | CNRM-CM6.1 | MPI-ESM1.2 | CMCC-CM2-(V)HR4 |
| Resolution names | LM, MM, HM | LR, HR | LR, HR | HR, XR | HR4, VHR4 |
| Model atmosphere component | MetUM | IFS cyc36r4 | ARPEGE6.3 | ECHAM6.3 | CAM4 |
| Atmospheric dynamical scheme (grid) | Grid point (SISL; lat-lon) | Spectral (linear; reduced Gaussian) | Spectral (linear; reduced Gaussian) | Spectral (triangular; Gaussian) | Grid point (finite volume; lat-lon) |
| Atmospheric grid name | N96; N216; N512 | Tl255; Tl511 | Tl127;Tl359 | T127; T255 | 1°×1°; 0.25°×0.25° |
| Atmospheric mesh spacing at 50ºN (km) | 135; 60; 25 | 71; 36 | 142; 50 | 67; 34 | 64;18 |
| Atmospheric nominal resolution (CMIP6) | 250; 100; 50 | 100; 50 | 250; 50 | 100; 50 | 100; 25 |
| Ensemble members at each resolution | 5; 3; 3 | 2*; 2* | 1; 1 | 1; 1 | 1; 1 |
| Atmospheric model levels (top) | 85 (85km) | 91 (0.01 hPa) | 91 (78.4 km) | 95 (0.01hPa) | 26 (2hPa) |
| Reference(s) | Williams et al. (2018); Roberts et al. (2019) | Haarsma et al. (2020) | Voldaire et al. (2019) | Gutjahr et al. (2019) | Cherchi et al. (2019) |

**Table 1: Summary of PRIMAVERA models used for the event set. *Only 2 of the 3 ensemble members run at each resolution were used due to tracking failures.**


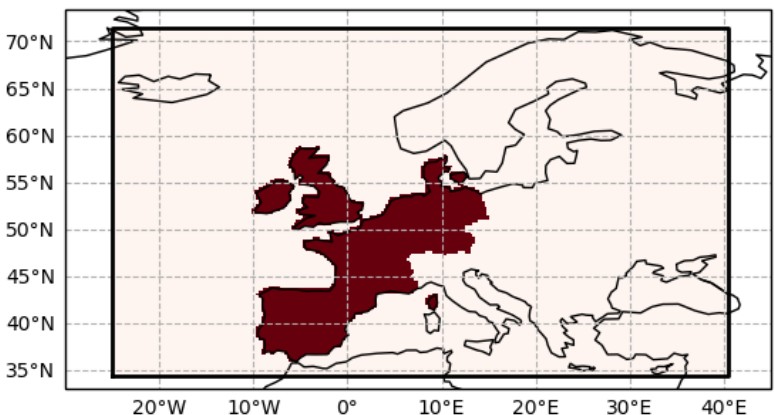

**Figure 1: Footprint domain and countries used for calculation of loss index.**


Footprints were made for every extra-tropical cyclone track identified by the TRACK algorithm. Many of these cyclones do not have strong enough winds to cause damage, but since users will be interested in different domains and use different estimations of storm severity or vulnerability functions, all footprints are retained so that users can perform filtering tailored to their own needs.

### 3.1.1 Storm tracking


The identification and tracking of the extra-tropical cyclones in the model data is performed following the approach used in Hoskins and Hodges (2002) based on the Hodges (1995, 1999) tracking algorithm (TRACK). The cyclones are tracked on the 6-hourly, T42 spectrally filtered 850 hPa relative vorticity field. Planetary waves with a wave number less than 5 are filtered out to remove the large-scale background and improve reliability of the algorithm. Only cyclones with a maximum intensity greater than $1.0\times10^{-5}$ s$^{-1}$ lasting at least 2 days and travelling more than 1000 km are retained for the footprints.


The tracking was performed on individual seasons (DJF, MAM, JJA, SON), but footprints were generated for all cyclones identified in the extended winter (October–March). Some cyclone tracks may have been cut short by crossing the season boundaries, or split into two separate tracks, but assuming a constant cyclone formation rate throughout the season and that the severe winds of a cyclone last 72 hours (as is assumed by in the insurance industry), this will only affect 3% of tracks.


The storm tracking in the ERA-Interim re-analysis was performed in a similar way (see Roberts et al., 2014 for a full description). The main difference is that the re-analysis tracking used 3-hourly data, but only 6 hourly track positions were retained to be consistent with the PRIMAVERA model tracks.

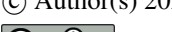



Due to technical issues, occasionally the tracking algorithm was unable to complete a full winter season. It was not possible to diagnose the issue and re-run the tracks in the time frame available, so these seasons were removed from the event set. 12 model winters were affected, listed in Appendix A.

### 3.1.2 Footprint separation

It was not possible to apply the method of footprint separation used in previous studies such as Roberts et al. (2014), since this
was developed for 6-hourly maximum wind data rather than daily maxima. Instead, to separate footprints of storms with overlapping 72-hour periods, for each day in a model run, each grid point in the daily maximum wind field is assigned to a storm track, by identifying the closest cyclone track point during that day. Grid points more than 1500 km from any track point are not assigned to a cyclone track. To generate the footprint for each cyclone track, the daily maximum winds for the 72-hour period (specified as above) are extracted, with the grid points assigned to other cyclones masked out, and the 72-hour
maximum is taken. Figure 2 demonstrates the method in ERA5 data for the observed famous storms Lothar and Martin, which struck France within 24 hours of each other on 26th–27th December 1999. Without separating the wind fields in this way, Fig. 2 shows that the footprints of Lothar and Martin would be almost identical over land.

On some occasions storm tracks can come within 1500 km of each other in the same 24-hour period, making it impossible to
separate the storms using daily data. In these cases the winds can be assigned to the wrong storm and the footprints appear to have truncated wind fields (this can be seen over the Atlantic in the final footprint for storm Martin, Fig. 2(h)). However, inspection of footprints in the event set shows that most strong winds (>20 ms$^{-1}$) over land are captured in each footprint, and the truncation should not have an effect on seasonal aggregate losses.

### 3.1.3 Downscaling and bias correction/conversion from winds to gusts

Insurance industry windstorm footprints are typically maps of maximum 3-s gust at 10 m rather than windspeed, on a very high resolution grid (maximum grid spacing ~25 km, although <10 km is preferred; Bojovic et al., 2017). To be consistent with these industry standards, the footprints must be converted from wind to gust speeds and downscaled to a common grid. Here the target grid is that of the ERA5 gusts (0.25°×0.25°, approximately 18 km grid spacing at 50° N), and ERA5 gusts are taken to represent observations.


We use quantile mapping to achieve both the conversion from winds to gusts and bias-correction. The method is as follows: The model daily maximum wind speeds are downscaled to the ERA5 grid using linear interpolation. At each *individual grid point*, the empirical cumulative distribution function (CDF) is calculated at probability intervals of 0.5% up to the 98th percentile. Above the 98th percentile, the CDF is fitted using a generalised Pareto distribution (GPD), which is commonly
used for fitting extreme wind speeds (e.g., Sharkey et al., 2020). Following Fawcett and Walshaw (2012), declustering to remove temporal dependence is not applied to improve precision of parameter estimates. The mean extremal index, estimated



using intervals estimator of Ferro and Segers (2003), has mean value 0.61, so the effect of clustering on the return levels is expected to be small relative to other errors (Fawcett and Walshaw, 2012). The GPD fitting is performed separately for each model.

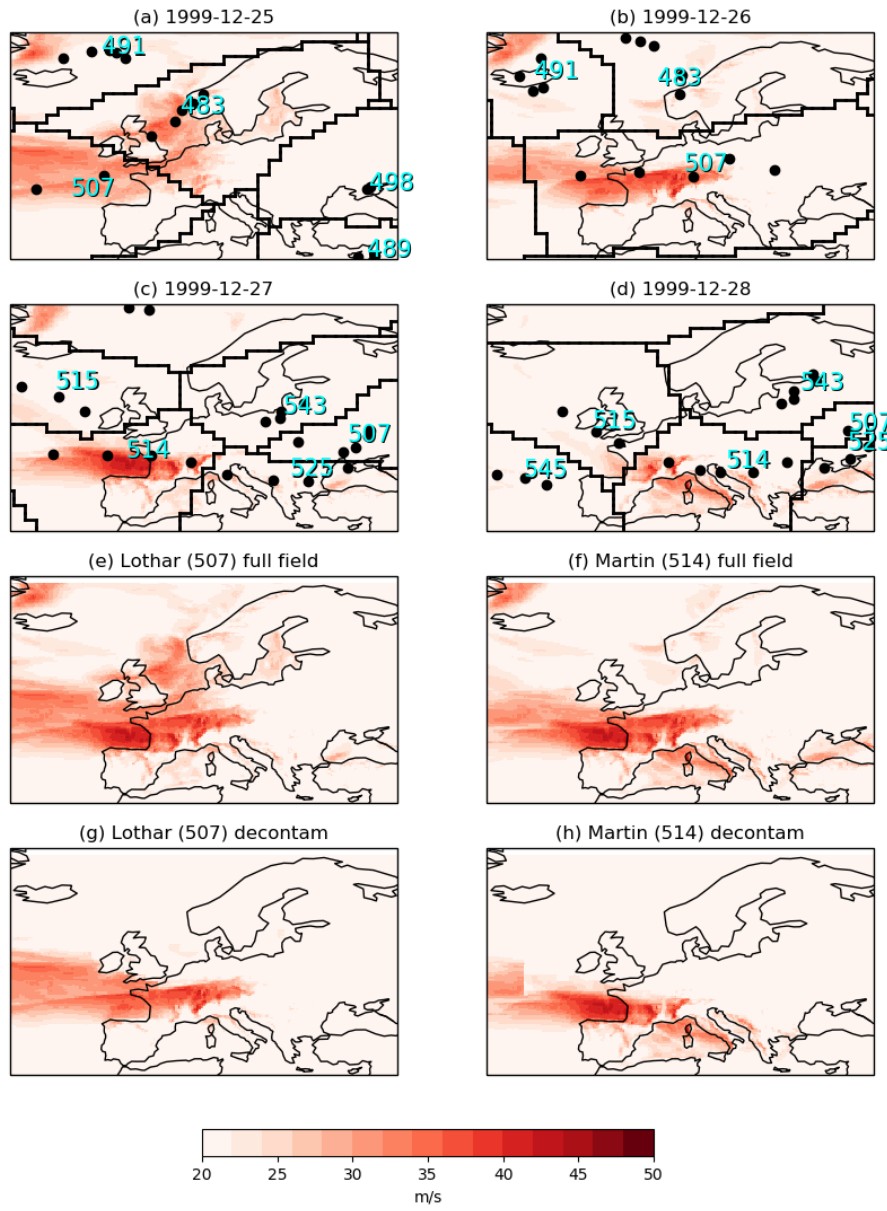

**Figure 2: Storm separation method: (a), (b), (c) and (d) show the daily maximum gust fields from ERA5 for 25th–28th December 1999, with the 6 hourly track points of each storm identified in the domain on that day (the cyan number gives the track identification number). The thick black lines mark the shape of the mask used for each track on each day; for example, the footprint of Lothar (track 507) is made by taking the maximum of the gusts in the marked areas around track 507 (the rest of the gusts are set to missing data) for 25th, 26th and 27th December. The resulting footprint is shown in panel (g), compared to taking the maximum gusts over the whole domain on the same days shown in panel (e). (f) and (h) are the same for storm Martin (track 514).**





The quality of the GPD fits was assessed by calculating the difference between the fitted and empirical value of the 99.8[th] percentile. If this was found to be greater than 1 ms[-1] at a grid point $i$, then the parameters of the GPD fit were taken from the

mean of the surrounding grid points. The CDF estimations as described above were then repeated on the ERA5 gust distribution. The model CDFs are estimated on wind speeds in the time period which overlaps with the ERA5 dataset, 1979/80–2013/14 (October–March only), to take into account any non-stationarity in the wind/gust speed distribution due to climate change and/or low frequency climate variability.

The daily maximum gust speeds, $g_i(t)$, at each grid point $i$ and time $t$ are then estimated using a transfer function:

$$g_i(t) = f^{-1}_{\text{ERA5},i}[f_{\text{mod},i}(w_i(t))], \tag{1}$$

where $w_i(t)$ is the daily maximum model wind speed at grid point $i$, and $f_{\text{ERA5},i}(x)$ and $f_{\text{mod},i}(x)$ are the estimated CDFs of the ERA5 gusts and model windspeeds at grid point $i$ respectively.

Quantile mapping has been used for this purpose in previous event set methodologies (e.g., Steptoe, 2017; Osinski et al., 2016), but note that here quantile mapping is performed for each grid point individually rather than pooling data over the whole domain. The reason for this is demonstrated in Fig. 3, which shows quantile-quantile (q-q) plots of the October–March daily maximum gusts from ERA5 against the October–March daily maximum wind speeds from the PRIMAVERA model HadGEM3-GC3.1-MM (other models are shown in Appendix B), for a selection of major cities around Europe. The mapping

from winds to gusts varies considerably depending on location, e.g., a 5 ms[-1] wind speed in London maps to a gust speed of ~9 ms[-1], whereas the same wind speed in Geneva maps to a gust speed of nearly 21 ms[-1]. One of the reasons for this discrepancy is the use of an effective roughness parametrisation in climate models, to take into account the effects of sub-grid scale orography and simulate realistic orographic drag on the upper level flow (Wood and Mason, 1993; Howard and Clark, 2007; Williams et al., 2020). This can, however, lead to unrealistically low surface wind speeds, especially over high land (Roberts

et al., 2014), and the degree of the bias is strongly dependent on the orographic properties of the grid point.

A disadvantage with mapping on individual grid points is that there are a limited number of data points available for fitting the GPD, so extreme gust values in the corrected footprints should be considered highly uncertain. In some cases, the mapping from model winds to gusts becomes unstable leading to unrealistically high estimated gusts, for example when an input model

wind speed is greater than the maximum used in the fitting period. In these cases the estimated gusts are capped at 60 ms[-1] over land and 70 ms[-1] over sea grid points. The limitations of this method to convert wind speeds to gusts are discussed further in Sect. 5.


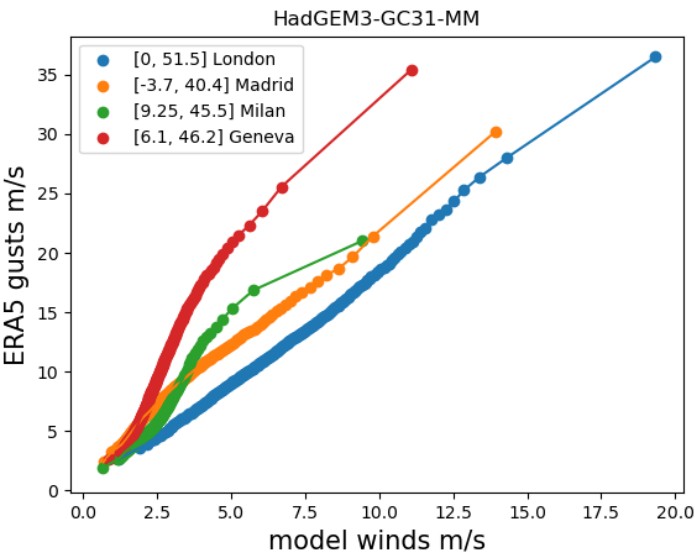

**Figure 3: q-q plots at selected locations (points in increments of 0.5%) showing relationship between ERA5 daily maximum 3-s gusts and daily maximum model winds in HadGEM3-GC31-MM. The PRIMAVERA wind speeds have been linearly interpolated to the ERA5 grid.**

### 3.2 Loss estimation

To estimate the damage resulting from each storm in the event set, for each storm footprint we calculate a dimensionless loss index (LI), based on the index derived by Klawa and Ulbrich (2003):

$$LI = \sum_i area_i \times pop\ dens_i \times \left(\frac{v_i}{v_{98,i}} - 1\right)^3 \text{ for } v_i > v_{98,i} \qquad (2)$$

where *area$_i$* and *pop dens$_i$* are the area and population density of grid point $i$, $v_i$ is the maximum gust speed in the footprint at grid point $i$, and $v_{98,i}$ is the 98th percentile gust speed at that same location. Following the approach of the WISC project

(Steptoe, 2017), unless stated otherwise, the area summation is over land grid points for the following countries: Luxembourg, United Kingdom, Ireland, France, Spain, Portugal, Belgium, Netherlands, Germany, and Denmark (shaded countries in Fig. 1).

This loss index has been found to correlate well with aggregate insured losses over a season in Germany and the UK (Klawa

and Ulbrich, 2003; Leckebusch et al., 2007). The index is calculated per event rather than the summation of daily data over a season, which is a commonly used modification (e.g. Karremann et al., 2014; Priestley et al., 2018). For the aggregate (total) losses per season, the LI is summed over all *events* in a season.

### 3.3 NAO calculation

In Sect. 4.4 the NAO index is defined as the anomaly in the difference between mean sea level pressure between a region

centred on the Azores (longitude 20 to 28° W, latitude 36 to 40° N) and one centred on Iceland (longitude 16 to 25° W, latitude





63 to 70° N; Dunstone et al., 2016). In this paper the extended winter mean NAO is calculated for the re-analysis and each climate model. The anomalies each winter are given with respect to the extended winter mean of the whole period available for each model (1950/51–2013/14) and the re-analysis (1979/80–2013/14), although almost identical results are obtained when anomalies are given with respect to the common period.

## 4 Results

### 4.1 Storm tracks and footprints

Footprints were generated for all extended winter tracks identified by TRACK for all models, producing a total of 1332 years of data. In total there are 268 620 footprints, 69 482 of which have a non-zero loss index (LI), and 2 738 represent severe damage storms (defined as those with LI $> 1.0 \times 10^6$, which occur approximately once every two winters over Europe and make up 70% of total losses). Table 2 compares the mean number of storms per extended winter in PRIMAVERA models to re-analysis, for all storms, storms with a non-zero LI, and severe storms. Numbers compare well with re-analysis, although all PRIMAVERA models appear to slightly underestimate the total number of storms. The number of footprints with a non-zero LI tends to increase with model resolution, possibly because to have LI>0 there must be regions with wind speeds greater than the local 98$^{th}$ percentile, which may occur in a higher proportion of storms if small scale features embedded with high wind speeds are better resolved. The mean number of severe storms per winter remains remarkably stable at approximately two storms per winter, matching the re-analysis.

The increase in storm numbers with resolution is also reflected in Fig. 4, which shows the track densities for footprints which have non-zero LI. The maximum track density is located over the UK, which is expected given the area used to calculate the LI (Fig. 1), and the fact that maximum winds tend to occur south of the tracks. The underestimation of non-zero LI tracks is most pronounced over the UK and the western parts of the European continent, but the bias is much reduced in the higher resolution models.

Figure 5 shows a selection of some of the most damaging storms from the re-analysis and ones of similar strength (as measured by the LI) from the PRIMAVERA models. The figure shows that the models can simulate different 'types' of storms, for example a large area storm like Daria (January 1990); intense, narrow storms such as Anatol (December 1999); storms with a southern track hitting the Iberian peninsula, such as Klaus (January 2009); and storms with a strong southwest-northeast tilt which travel northwards from Iberia to northern Europe such as 87J (the Great Storm of 1987; October 1987). Note that the model simulations are not attempting to simulate the re-analysis storms (as can be seen by the very different dates for the footprints), the figure is simply to illustrate the variety of storms that can be simulated.





| Names of the low-, med- and high-resolution models | Mean number of footprints per winter | | | Mean number of footprints with non-zero loss index per winter | | | Mean number of severe footprints per winter | | |
|---|---|---|---|---|---|---|---|---|---|
| | Low res | Med res | High res | Low res | Med res | High res | Low res | Med res | High res |
| CMCC-CM2-HR4 CMCC-CM2-VHR4 | 215.3 | | 220.6 | 42.1 | | 62.8 | 2.3 | | 2.2 |
| CNRM-CM6-1 CNRM-CM6-HR | 198.3 | | 216.1 | 44.3 | | 61.9 | 2.1 | | 2.3 |
| EC-Earth3P EC-Earth3P-HR | 200.6 | | 199.3 | 52.6 | | 66.8 | 2.2 | | 2.3 |
| HadGEM3-GC3.1-LM HadGEM3-GC3.1-MM HadGEM3-GC3.1-HM | 189.9 | 202.6 | 205.96 | 41.6 | 51.5 | 61.7 | 2.0 | 1.8 | 1.8 |
| MPI-ESM1.2-HR MPI-ESM1.2-XR | 203.6 | | 206.2 | 45.1 | | 54.4 | 2.2 | | 2.4 |
| **Low res models, high res models*** | **196.8** | **206.1** | | **44.4** | **59.2** | | **2.1** | **2.0** | |
| **All models** | **201.7** | | | **52.2** | | | **2.1** | | |
| **Re-analysis** | **237.8** | | | **64.7** | | | **2.0** | | |

**Table 2: Mean number of total footprints, footprints with a non-zero loss index and severe footprints per winter for each PRIMAVERA and re-analysis footprints. For each PRIMAVERA model, the means from both the low and high-resolution versions are given in each row. *The model HadGEM3-GC3.1-MM is included in the 'high-res' count.**

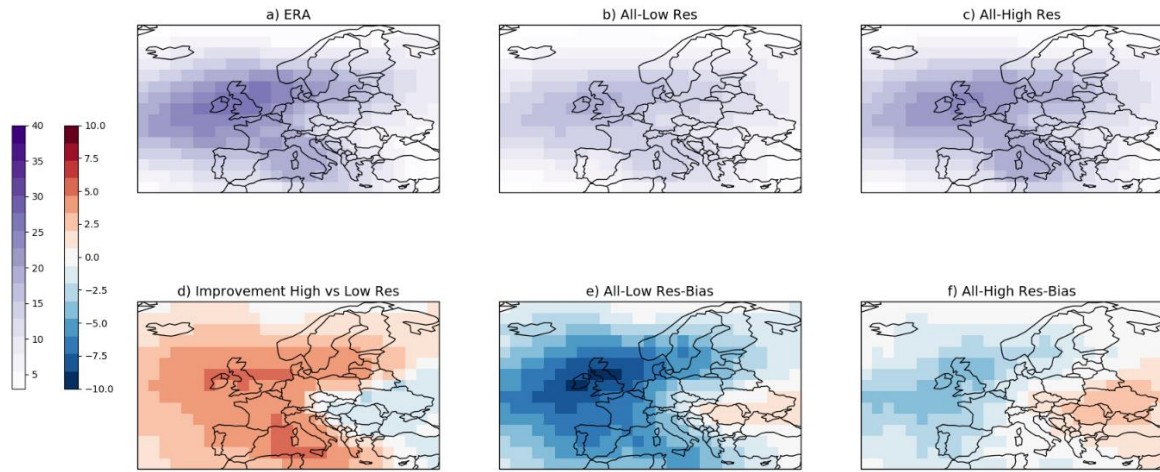

**Figure 4: Track densities from re-analysis (a) and PRIMAVERA models ((b) and (c)), for tracks with a non-zero loss index. Following Economou et al., (2015), track density is defined by the number of tracks with at least one track point passing within 6.3° of each grid box (on a 2.5°×2.5° grid) per winter. The track density and bias (model – re-analysis) for the low resolution PRIMAVERA models are shown in panels (b) and (e), and for the higher resolution PRIMAVERA models in panels (c) and (f). The medium resolution version of HadGEM3-GC3 is included in the higher resolution models. The change in bias (|low resolution bias|-|high resolution bias|) from low to high resolution is shown in panel (d), with red areas corresponding to improvement with increased resolution.**

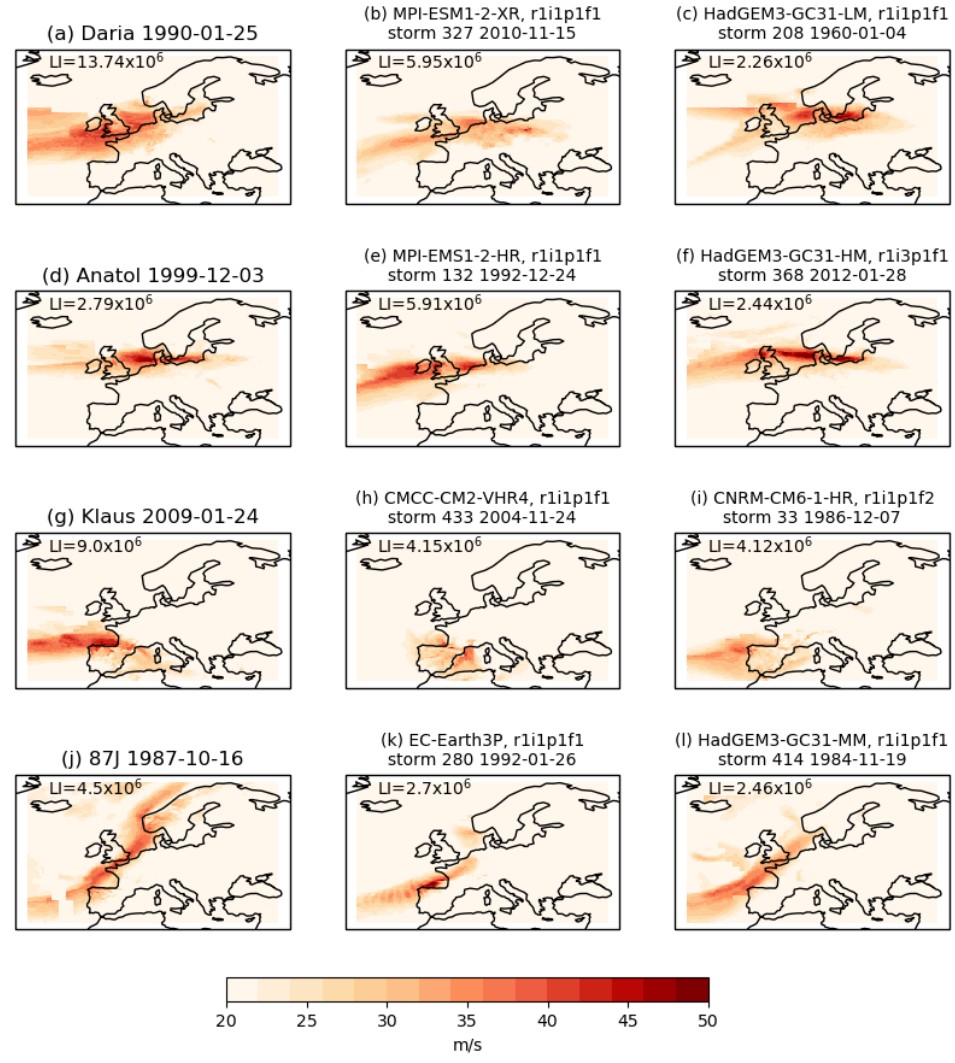

**Figure 5: Example observed and simulated footprints: Left column (a), (d), (g), (j) are re-analysis footprints of the famous storms Daria, Anatol, Klaus and 87J (The Great Storm of 1987). The footprints in the middle and right columns are from PRIMAVERA models, with each row showing two examples of storms of a similar type to the re-analysis examples. The central date of each footprint is given in the panel titles.**

## 4.2 Loss index distribution

We now examine how well the PRIMAVERA models capture the intensity distribution of storms, as measured by the loss index. Figure 6 plots the distribution of number of severe storms (those with $LI>1\times10^6$) per extended winter as a function of LI for PRIMAVERA and re-analysis data. The re-analysis data set contains only 35 winters so the distribution contains more noise than that for the PRIMAVERA models. For a fair comparison with the models, we therefore take 1000 random samples (with replacement) of 35 winters from the PRIMAVERA data set to estimate the noise in a 35-winter sample. The vertical red



lines on the PRIMAVERA distribution in Fig. 6 show the 95% interval for each intensity bin based on the random samples. The re-analysis distribution lies well within the sample distributions of the models, showing that the models' and re-analysis distributions are consistent with one another.

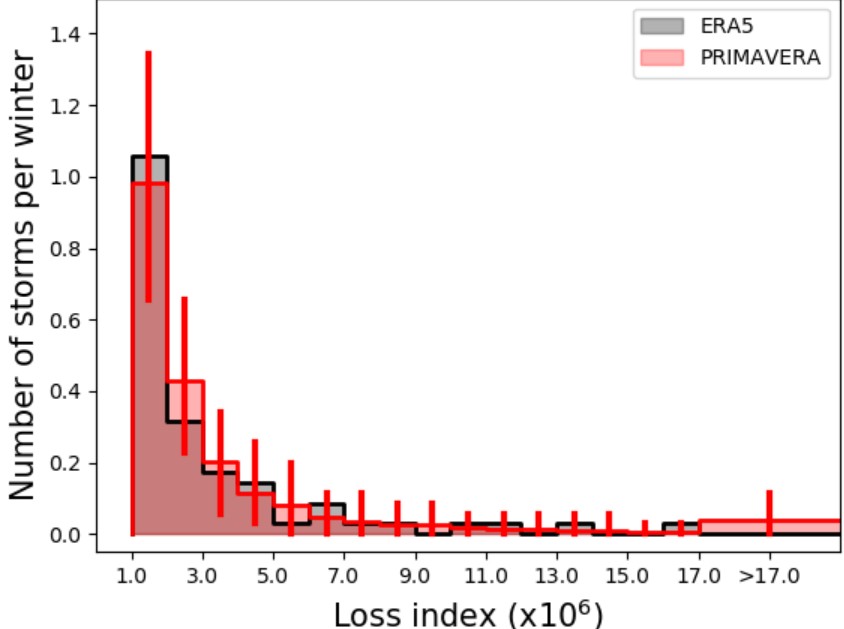

**Figure 6: Distribution of number of severe storms (LI>1×10⁶) per extended winter as a function of LI in PRIMAVERA models (red)**
**and re-analysis (black). Vertical red lines show the 95% range in frequency estimated from 1000 35 year samples (with replacement) from the model data. Note that the last LI bin (LI>17×10⁶) is larger.**

Figure 7(a) shows the return period curve for seasonal *aggregate* losses (seasonal sum of LI) in the PRIMAVERA models and the re-analysis data. The extreme tail of the PRIMAVERA data (seasons with an aggregate loss above the 90[th] percentile) is
fitted with a GPD curve (Welker et al., 2020, Walz and Leckebusch, 2019). Note that three model storms (listed in Appendix C) had to be removed from the seasonal aggregate losses as they were considered unrealistically extreme, and their inclusion prevented a satisfactory GPD fit. The aggregate losses before their removal are shown with the open red circles in Fig. 7(a). This is discussed further in Sect. 5.

The 95% confidence intervals on the GPD fit have been quantified by repeatedly (1000 times) randomly sampling M years of data from the fitted function (where M is the number of years of data used in the original fit, equal to 1332 for PRIMAVERA) and then re-fitting. The GPD fit estimates that the most extreme season over Europe in re-analysis (1989/90), which had a total LI of $4.5×10^7$, has a return period of 75-200 years under present day conditions, which is much longer than the 35 years estimated from the re-analysis data alone.


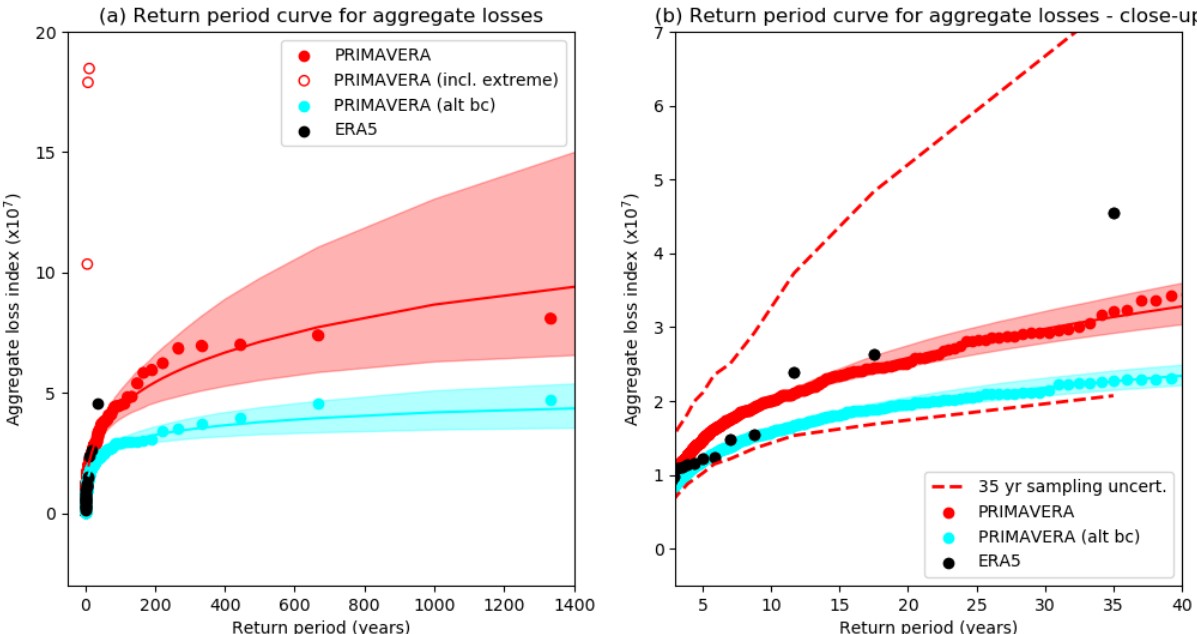

**Figure 7: Return period curve for seasonal aggregate losses. The GPD fit to PRIMAVERA data is shown by the red line, with individual seasons shown by the red dots. The open red circles show the aggregate losses for the three seasons which contained the unrealistically extreme storms, before these storms were removed from the aggregate. The cyan points show the losses from the PRIMAVERA data using an alternative bias correction/gust conversion method described in Sect. 5. The shading shows the 95% confidence intervals to the GPD fits, estimated by re-sampling. ERA5 data is shown in black. Panel (b) is a close-up version of panel (a). The dashed lines in (b) show the 95% confidence intervals in LI for a given return period when sampling 35 years of PRIMAVERA data.**

To check if the PRIMAVERA models' return periods are consistent with the re-analysis data, as in Fig. 6, 1000 35 year samples were taken from the PRIMAVERA dataset, and the 95% intervals of the measured aggregate losses for each return period are shown with the dashed lines in Fig. 7(b). The ERA5 losses are well within the bounds of the PRIMAVERA data and demonstrate the huge uncertainty in losses/return periods when only 35 years of data are used. Figure 7 also shows the return period curve for PRIMAVERA aggregate losses when an alternative bias correction/conversion to gusts is used, which is discussed further in Sect. 5.

### 4.3 Storm clustering

Serial (or temporal) clustering of windstorms is the tendency of these events to arrive in groups (Dacre and Pinto, 2020). It has been shown in both observations and climate models that storms are serially clustered in the flanks and exit region of the North Atlantic storm track, and thus on their arrival into Europe (e.g. Mailier et al., 2006; Vitolo et al., 2009; Pinto et al., 2013; Economou et al., 2015). Priestley et al. (2018) demonstrated the importance of clustering in estimating losses from a high-





resolution climate model, with seasonally aggregated losses 20% higher in the (clustered) climate model output compared to random re-sampling of the data assuming a Poisson distribution for the storm frequency.

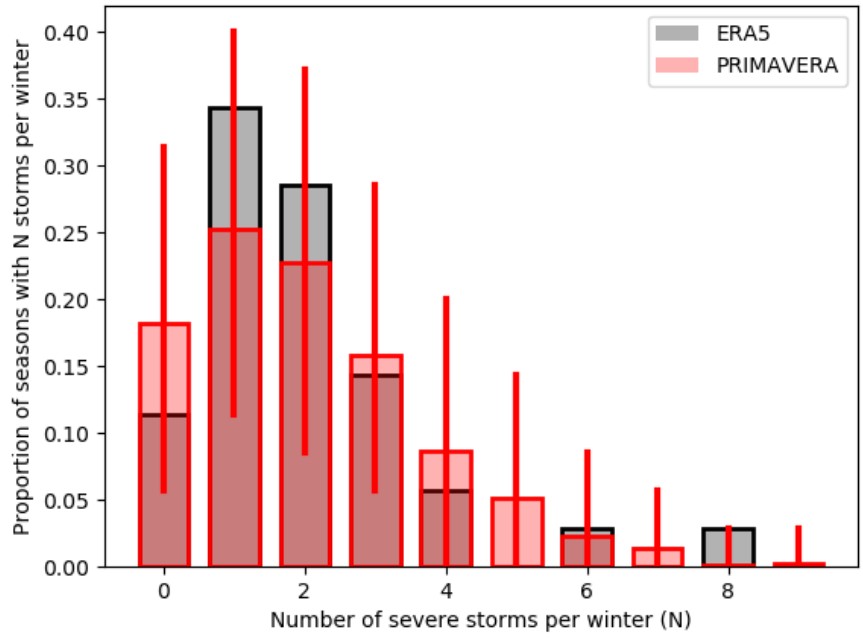

**Figure 8: Distribution of number of severe storms per winter in PRIMAVERA models (red) and re-analysis (black). The red lines on each bar show the 95% range of season counts for 1000 35 re-sampled years of PRIMAVERA data.**

We assess clustering in the PRIMAVERA simulations by comparing the distribution of the frequency of severe storms ($LI>1\times10^6$) per season to re-analysis (Fig. 8). As in Fig. 6, the consistency between PRIMAVERA and re-analysis data is assessed by taking 1000 35 year samples of PRIMAVERA data. The re-analysis distribution is consistent with being a sample from the PRIMAVERA data, and the models can even simulate seasons as extreme as winter 1989/90, with 8 severe storms over Europe. There are 5 (of 1332) seasons in the model with ≥8 severe storms, giving the chance of this occurring at least once in a 35 year period of $1-(1327/1332)^{35}=12\%$.

The PRIMAVERA storm numbers show a clustered distribution, with the dispersion statistic (equal to $\sigma^2/\mu$ - 1, where $\sigma^2$ is the variance of storm counts per season and $\mu$ is the mean; Mailier et al., 2006) of 0.38, which is significantly greater than zero with 95% confidence (p=0.018, estimated from the distribution of dispersion statistic assuming a Poisson distribution), and close to the re-analysis value of 0.35 (significantly greater than zero with 90% confidence, p=0.07).


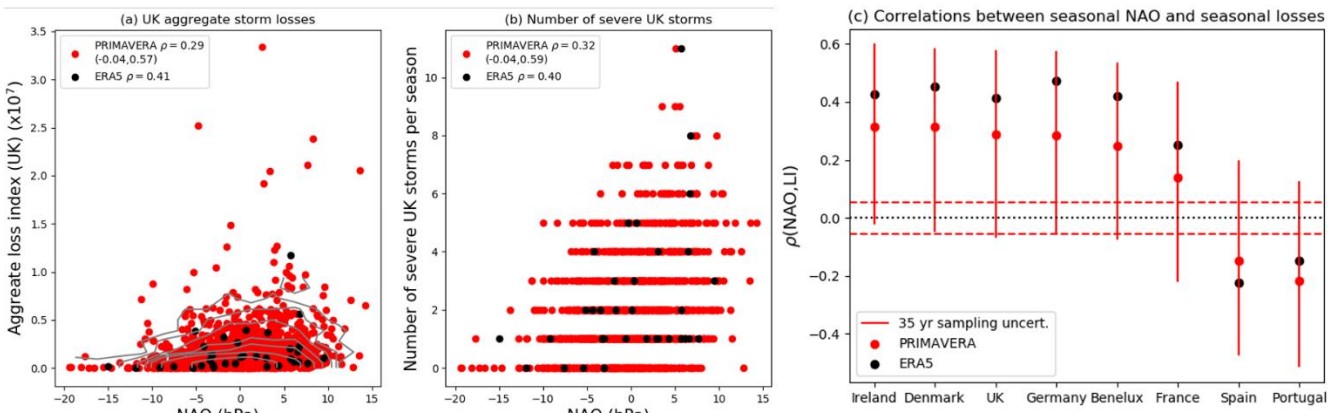

**Figure 9: The relationship between extended winter NAO and storm damage. (a) Scatter plot of extended winter aggregate loss over the UK against extended winter NAO for PRIMAVERA and re-analysis data. Contour levels (for aggregate losses < 1×10⁷) are shown to illustrate the density of PRIMAVERA data. The contour levels are 2, 5, 10, 20, 30, 50 and 100 seasons, in bins of width 2.5 hPa in NAO, and 1.1×10⁶ in aggregate losses. The rank correlation coefficients between aggregate losses and NAO are given in the legend. For PRIMAVERA data, the 95% range of correlation coefficients when taking 1000x35 year random samples is also shown in brackets. (b) As in (a) but showing the number of severe UK storms (UK LI > 1×10⁵) per season. (c) Rank correlation coefficients between seasonal aggregate LI and NAO over the countries in the European domain for PRIMAVERA (red dots) and re-analysis (black dots). The vertical red solid lines indicate the 95% distribution of correlations from 1000 35 year samples from PRIMAVERA data (not the confidence intervals on the correlation coefficient of all 1332 years of data), to show consistency with re-analysis. The red dashed lines show the 95% confidence intervals of correlation coefficients for 1332 years uncorrelated data: PRIMAVERA correlations are all outside these intervals indicating a significant difference from zero correlation with at least 95% confidence.**

## 4.4 Dependence on NAO

The North Atlantic Oscillation (NAO) is the primary mode of variability in the North Atlantic European region (Wallace and Gutzler, 1981), and is closely linked with the position of the North Atlantic storm track (Rogers, 1990), and consequently European windstorm damage (Walz et al., 2020). Recent advances in the predictability of the NAO on timescales of seasons to decades (Scaife et al., 2014; Dunstone et al., 2016; Athanasiadis et al., 2020; Smith et al., 2020) have thus opened up the possibility of being able to predict European storminess on long range timescales (Befort et al., 2018). Since the footprints in PRIMAVERA are generated from a global climate model, it is possible for insurers to extract storms associated with different NAO states to estimate the effect of the NAO on their particular portfolio, and even estimate the change in expected losses for a given NAO forecast.

Figure 9 shows the extended winter aggregate UK LIs and (b) severe storm counts over the UK against extended winter mean NAO, from PRIMAVERA models and re-analysis. The threshold for severe storms is reduced from the European LI value of 1×10⁶ to 1×10⁵, to take into account the smaller area and population of the UK. The UK is chosen as an example because it is well within the northern region of influence of the NAO (e.g. Hurrell and Deser, 2010).





Figure 9 (a) shows there is a non-linear relationship between aggregate storm loss and NAO, with a clear increase in risk of a
420    high loss season as NAO increases. Although the rank correlation coefficients ($\rho$) between aggregate losses and NAO for
PRIMAVERA and re-analysis are modest (0.29 and 0.41 respectively, both statistically significantly different from zero with
>95% confidence), PRIMAVERA data estimates that the probability of an extreme season over the UK (defined as having an
aggregate seasonal loss above the 90th percentile, $4\times10^6$) increases to 0.2 for NAO>5 hPa, and decreases to just 0.06 for NAO<-
5 hPa.


There is a similar positive correlation for the *number* of severe storms striking the UK each winter ($\rho$=0.32 for PRIMAVERA
and 0.40 for re-analysis, see Fig. 9(b)). Figure 9 (c) shows the correlations between extended winter NAO and aggregate
losses for the other countries in the domain and shows the expected relationship with positive (negative) correlations for the
northern (southern) countries. The 95% significance levels for the PRIMAVERA data (from a 2-tailed t-test) are shown by
the dashed red lines, indicating that all the PRIMAVERA correlations (shown by the red dots) are statistically significant. As
before, consistency with ERA5 is tested by randomly sampling 35 year time series from the PRIMAVERA dataset, and the
95% range of correlations for 35 year samples are shown by the solid red vertical lines in Fig. 9(c). All the ERA5 correlations
are within the bounds of the PRIMAVERA data.

## 5 Uncertainty in storm severity due to the bias correction/conversion to gusts method

Figure 7 showed that there were 3 model storms with unrealistically high loss indices, which had to be removed from the
seasonal aggregate losses to obtain a satisfactory GPD fit for the calculation of return periods. This is due to the quantile
mapping method to bias correct/convert to gusts, which relied on GPD fits of daily PRIMAVERA model and ERA5
winds/gusts (Sect. 3.1.3). As the GPD fitting is performed separately for each model over the common time period with ERA5,
and at individual grid points, only 35 years of data are available for each fit (apart from where multiple ensemble members are
available). Inspection of the most extreme footprint shows an intense storm centred over Barcelona, where the input raw model
winds in this region were substantially greater than the maximum model winds used in the GPD fitting period. Due to the high
sensitivity of the transfer functions, and the fact that the loss index is population weighted and dependent on the cube of the
gust, the high gusts in this area have a huge impact on the resulting loss index.

In fact, all of the 10 most intense model storms (as measured by the LI) were for storms outside the fitting period used for the
bias correction, indicating large uncertainty in the maximum gusts possible at each grid point. In addition, 9 of the 10 most
intense model storms are centred on southern Europe, off the main storm track, where excessive gusts will have a larger impact
on the LI due to the lower 98th local gust percentile. 9 of the 10 most intense storms are also produced from the lower resolution
version of each model, which may indicate issues in the transfer functions when there is a large change in resolution from
native to target grid.

Natural Hazards
and Earth System
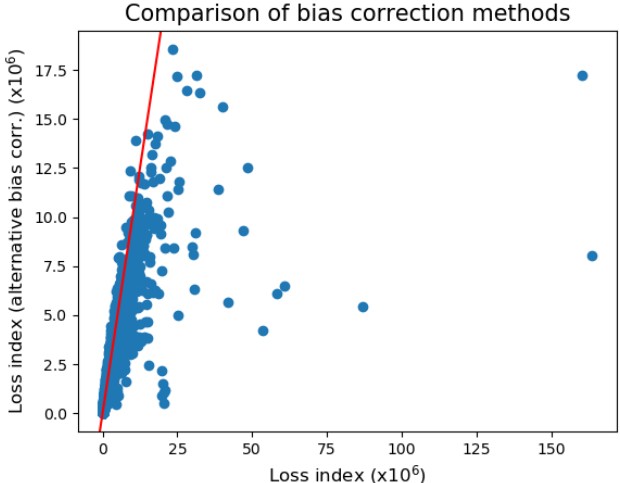

**Figure 10: Scatter plot of European LI for individual footprints from the alternative bias correction method (Sect. 5) against LI using the original method. The red line shows equality.**

Therefore, to estimate the sensitivity of LI to the estimated cumulative distribution function of ERA5 gusts and model winds, we tested using an empirical quantile mapping method (e.g. Steptoe, 2017, Osinski et al., 2016) to convert from winds to gusts. Here, instead of fitting the CDF of the gusts/winds with a GPD curve, we linearly interpolate between the empirically estimated quantiles. When an input model wind is greater than the maximum wind used in the fitting period, it is converted to the maximum ERA5 gust in the fitting period at that grid point.


Figure 10 shows a scatter plot of the loss index of each storm with this alternative bias correction method against the original loss index. There is strong agreement between the two methods, although the original method tends to give higher intensities. A few (~20) storms show a large discrepancy, with substantially higher LIs using the original bias correction method. The estimations of LI for these storms (when calculated over the countries shown in Fig. 1) should be considered unreliable.


Figure 7 shows the return period curve for the aggregate losses with the alternative bias correction method (plotted in cyan). Figure 7 (b) shows that this curve starts to diverge from the original dataset around the 5 year return period. It is likely that the alternative bias correction method gives an underestimation of the 'true' losses, since maximum gusts cannot be greater than those in ERA5. Nevertheless, it demonstrates the sensitivity of loss estimations for long return periods to the bias
correction method used.

Other bias correction methods include correcting for the effective roughness parametrisation which leads to the underestimation of model winds, as described in Howard and Clark (2007) (or a simplified version in Haylock, 2011), or pooling data for the GPD fits and allowing for dependencies on covariates such as altitude, roughness and latitude (see eg.
Economou et al., 2014, who pooled mean sea level pressure data from North Atlantic storm tracks to fit GPD functions, but

included dependence on latitude and NAO in the fit parameters). Alternatively, the relationship between winds on the native grid and high resolution gusts can be modelled (for example using linear regression) if there are like-for-like footprints on both grids. This was not possible here since PRIMAVERA models are free-running and not attempting to simulate individual storms in ERA5, but this could be achieved by dynamically downscaling a selection of model footprints, as in Haas and Pinto (2012).

## 6 Conclusions

We have produced a freely available winter windstorm event set from PRIMAVERA global climate models for use in insurance risk analysis, which consists of 268 620 windstorm footprints, covering 1332 years of data. The method developed to create the event set separates the footprints of storms in the domain during overlapping time periods, allowing characteristics such as storm clustering to be studied more easily. To be consistent with the insurance industry definition of a footprint, the raw model winds were statistically converted to gusts on a $0.25° \times 0.25°$ grid. The intensities of the most severe storms in the event set are, however, sensitive to the gust conversion/bias correction method used.

The damage over Europe from each storm is estimated with a loss index. The frequency distribution of estimated European windstorm losses from the resulting event set, as well as the total losses per season, are consistent with re-analysis, and the additional event set data greatly reduces uncertainty on return period magnitudes. The event set also reproduces the distribution of the number of severe European storms per season seen in re-analysis, which is statistically distinct from a Poisson distribution and confirms the temporally clustered nature of severe European windstorms. The PRIMAVERA data suggest that the total loss of the most extreme season in the re-analysis data, winter 1989/90, has a return period of 75-200 years (in present day conditions), longer than the empirical estimation from re-analysis (35 years).

The model also simulates a relationship between extended winter aggregate storm loss and the extended winter mean NAO, consistent with the re-analysis data. Although only moderate (but statistically significant) positive correlations between seasonal NAO and aggregate losses are found for northern European countries, the probability of extreme losses in a season (>90th percentile) for the UK increase by a factor of 4 in positive NAO (NAO>5hPa) seasons compared to negative ones (NAO<-5hPa). Since monthly NAO values are provided with the dataset, this allows users to investigate the effect of NAO on their individual portfolios, and to quantify the impact of a given NAO forecast, opening the possibility of predictive catastrophe modelling.

Future work includes refining the conversion to gusts/bias correction method, and extending the event set to include the coupled PRIMAVERA simulations, and the PRIMAVERA climate projections which run to 2050.





# Appendices

## Appendix A

The following model winters were removed from the event set due to incomplete tracking:

- CMCC-CM2-VHR4_highresSST-present_r1i1p1f1: 1993/1994
- EC-Earth3P-HR_highresSST-present_r3i1p1f1: 1982/3 1983/4 1966/67 1967/68
- EC-Earth3P_highresSST-present_r1i1p1f1: 1950/51 1951/52
- EC-Earth3P_highresSST-present_r3i1p1f1: 1962/63 1963/64 1970/71 1971/72
- HadGEM3-GC31-HM_highresSST-present_r1i3p1f1: 2006/7





**Appendix B**



**Figure B1: As in Figure 3 but for the remaining PRIMAVERA models.**






## Appendix C

List of the 10 most extreme footprints in the event set according to European loss index. The first three storms were excluded from the seasonal aggregate losses when fitting the return period curve in Figure 7.

- CMCC-CM2-HR4_highresSST-present_r1i1p1f1_winter1957-1958_MAM_storm123_1958-03-24_regrid_corrected.nc, LI = $163.7\times10^6$
- EC-Earth3P_highresSST-present_r3i1p1f1_winter1954-1955_DJF_storm294_1955-01-29_regrid_corrected.nc, LI = $160.3\times10^6$
- CNRM-CM6-1-HR_highresSST-present_r1i1p1f2_winter1974-1975_SON_storm267_1974-11-05_regrid_corrected.nc, LI = $86.9\times10^6$
- HadGEM3-GC31-LM_highresSST-present_r1i2p1f1_winter1963-1964_SON_storm265_1963-10-25_regrid_corrected.nc, LI = $60.9\times10^6$
- CMCC-CM2-HR4_highresSST-present_r1i1p1f1_winter1978-1979_DJF_storm29_1978-12-06_regrid_corrected.nc, LI = $58.2\times10^6$
- CMCC-CM2-HR4_highresSST-present_r1i1p1f1_winter1968-1969_MAM_storm2_1969-03-
13_regrid_corrected.nc, LI = $53.5\times10^6$
- MPI-ESM1-2-HR_highresSST-present_r1i1p1f1_winter1976-1977_DJF_storm145_1976-12-29_regrid_corrected.nc, LI = $48.6\times10^6$
- HadGEM3-GC31-LM_highresSST-present_r1i14p1f1_winter1963-1964_DJF_storm240_1964-01-10_regrid_corrected.nc, LI = $46.9\times10^6$
- CMCC-CM2-HR4_highresSST-present_r1i1p1f1_winter1960-1961_MAM_storm12_1961-03-05_regrid_corrected.nc, LI = $41.8\times10^6$
- HadGEM3-GC31-LM_highresSST-present_r1i1p1f1_winter1958-1959_DJF_storm303_1959-01-25_regrid_corrected.nc, LI=$40.3\times10^6$

## Code Availability

The codes used to produce the results in this paper are available upon request to the contact author.

## Data availability

All raw PRIMAVERA model data is available from the Earth System Grid Federation (ESGF; https://esgf-index1.ceda.ac.uk/projects/esgf-ceda/). ERA5 and ERA Interim re-analysis data are available from the Copernicus Climate Data Store (CDS; https://cds.climate.copernicus.eu/#!/home). The final event set data can be accessed by emailing

julia.lockwood@metoffice.gov.uk.



**Author contribution**

EJP, MJR, HET, GSG and JFL were involved in conceptualisation of the work. JFL developed the footprint methodology. Analysis was performed by JFL and GSG. Storm track data was generated by MJR. SJB advised on the statistical analysis. AA provided insurance expertise. JFL prepared the manuscript with contributions from all co-authors.

**Competing interests**

The authors declare that they have no conflict of interest.

**Acknowledgements**

This paper is part of the IS-ENES3 project that has received funding from the European Union's Horizon 2020 Research and Innovation Programme under grant agreement No 824084. JFL, GSG, EJP and MJR acknowledge support by the
PRIMAVERA project of the Horizon 2020 Research and Innovation Programme, funded by the European Commission under Grant Agreement No 641727. SJB was supported by the Met Office Hadley Centre Climate Programme funded by BEIS and Defra. JFL would like to thank Kevin Hodges and Adrian Champion for kindly providing the ERA-Interim track data.

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
