# Peer review of "Using high-resolution global climate models from the PRIMAVERA project to create a European winter windstorm event set"

_Natural Hazards and Earth System Sciences, 2022_

## Referee Comment (RC2)

Review of *'Using high-resolution global climate models from the PRIMAVERA project to create a European winter windstorm event set'* by Lockwood et al.

**Overview**

This paper presents a windstorm footprint event set based on PRIMAVERA model simulations of the historical climate. This paper is very well written and the data described in this manuscript will be of great use both academically and for use in industrial sectors.

The authors present their footprint construction and calibration methods and also perform analysis as to how the NAO phase affects windstorm losses. Overall, I have only a few minor comments which should be addressed subject to acceptance and publication.

**Comments**

L112 – You have chosen to use the atmosphere-only simulations of the PRIMAVERA models. What is the justification of using these instead of fully coupled simulations?

L124-125 – For the models that output gust data, how does your final diagnostic compare to the model output gusts? Are they at all similar?

L289 – Why this threshold? Is it a particular threshold from the reanalysis LI or has it been arbitrarily chosen by the authors?

L298 and Fig. 4 – How does the track density of the severe storms compare to that of all the storms? As the severe storms are what you are interested in would this be more appropriate? Are the biases of the same magnitude and in the same locations?

Fig. 4 – are these biases significant?

L304-305 and Fig. 5 – this sentence needs re-phrasing/clarifying. In the caption it is clear you are showing footprints of similar structure (as far as I can tell). However the sentence states that you are showing those of similar LI, which is not the case. On this note i think it would also be very useful to have a comparison of similar LI storms as Daria, Kyrill, Anatol,etc.

L305 – How did you select these storms? Was this just done by eye or was there a quantitative measure to select them?

Table 2 and onwards – How does the inclusion of the MM HadGEM3 run affect the model bias? Does this overweight the means toward the HadGEM3 climate. Please quantify the impact of having both the HM and MM simulations compared to just having HM.

L345-347 – these storms that are discussed in Sect. 5 it would be good to show all the evidence behind removing these from the data. LI footprints could be included in the appendix.

L352-354 – This statement assumes that the distribution of LI is the same in the reanalysis as the models, which it may not be - it may also be good to do the GPD fit to the reanalysis data in Fig. 7b to compare to the fit of the reanalysis and models.

L387-391 – This dispersion value is this for cyclone counts in your entire European region? In Mailier et al. (2006) this is shown as a spatial field. Do your results compare spatially to this?

Fig. 9 – are figs 9a and 9b consistent for other geographic regions as is the case in Fig. 9b?

What is the impact of the 5 different models in your analysis? Do they all exhibit different behaviour? Are the results shown for example in Figs. 6, 8, 9 an artefact of the model mean or is this evident as a feature of each of the individual models?

You state the data is available from the author, however, I think this dataset would be very beneficial for general access. Are there plans to make this publicly available?

Some of the references quoted in the text are not included in the reference list – please double check the reference list.

---

## Author Comment (AC1)

**PRIMAVERA paper: Response to referees' comments**

We would like to thank all three referees for taking the time to read the paper and providing insightful and constructive comments. As some issues were picked up by all referees, we have put our responses into one document.

**Referee 1:**

**RC1.1:** I think that some hints on the requirements of the EIOPA, which wants EU based insurers to discuss their business in the context of climate change is missing

Thank you for bringing this to our attention. We will add it to the Introduction when discussing industry regulations:

*L39: "Insurance and re-insurance companies must estimate the financial risk posed by these events to ensure they are able to pay out the resulting claims**, and to satisfy industry regulations. For example, European law requires that EU based insurers hold enough capital to withstand the 1 in 200 year loss (Solvency II, 2009), and the European Insurance and Occupational Pensions Authority (EIOPA) states that insurers must discuss the impact of climate change on their business (EIOPA, 2022), which involves assessing trends in hazards in present and future climate."***

(We will remove the original sentence about Solvency II on line 46 to avoid repetition)

**RC1.2:** ERA5 wind gust data is set as a reference for the reality. I am missing a reality check or at least some references on the quality of the gust speed representation of the ERA5 data. Gust speeds are the main driver of the used loss function. If they are of minor quality the results will suffer from it, too.

Thank you for raising this important point, which has been brought up by all the referees. While there are many papers in the literature discussing the quality of wind speed climatology in ERA5, we were only able to find one paper on the validation of ERA5 gusts:

Minola, L., Zhang, F., Azorin-Molina, C. et al. Near-surface mean and gust wind speeds in ERA5 across Sweden: towards an improved gust parametrization. Clim Dyn 55, 887–907 (2020).

This study found that ERA5 gusts show an improvement compared to those in ERA-Interim, with high correlations with observed gusts in Sweden (r>~0.8, see their Fig 7). There is evidence of a negative bias for strong gusts, but since these mostly occur over mountainous regions we are unable to discern from this study whether the negative bias would affect *all* areas during strong gusts (e.g. during a wind storm), or if it is just a feature for mountainous regions.

We therefore conducted a quick comparison to station observations from data we had available from the Met Office Integrated Data Archive System (MIDAS) which were used in a previous study. For a selection of famous historical storms we have plotted the maximum observed 3-s gust (over a 72 hour period) against the ERA5 gust linearly interpolated to the station location, shown in Fig R1 below.

The results show reasonable agreement between ERA5 and observations, with data scattered around the *y=x* line and a clear positive correlation for all events (r=0.76 overall). We are therefore satisfied that ERA5 gives a reasonable representation of gusts.

We will add the following to Section 2.2 (L134):

*"A re-analysis was chosen to represent observations rather than station data because of complete spatial coverage, but we acknowledge that re-analyses can suffer from biases. There is limited literature on the validation of ERA5 gusts, although Minola et al. (2020) showed a high temporal correlation between ERA5 gusts and station observations in Sweden, with evidence of a negative bias for strong gusts over mountainous regions. We performed a comparison of the gusts in ERA5 footprints to station observations for a selection of 6 famous historical storms revealing reasonable agreement between ERA5 and observations (see [reference to this discussion])."*

**RC1.3:** The authors are using the output of different climate models, which probably differ in the representation of the storm tracks and other wind related issues. I would like to see some general figures on the representation of wind over Europe for each model in the appendix (e. g. storm tracks, high percentiles of wind, ...), . Perhaps some short remarks on the main differences compared to ERA5.

We have plotted the track densities of the non-zero loss index storms of individual models in Figure R3 below, which we will include in the Appendix of the manuscript. For this measure the models behave quite similarly, generally having too few storms over the UK and western Europe, with the bias improving for the higher resolution models. We have also plotted the track densities for intense storms only for comment RC2.4 for the multi-model means (Figure R2) and individual models (Figure R4), and plotted Figures 6, 8 and 9 for each model separately (see response to comment RC2.13), and again the general behaviour is consistent between models.

We will add to Section 4.1 (L303):

**"The track densities for the individual models are shown in Figures D1 and D2. The models all show the reduction in bias in non-zero LI storm numbers over western Europe as resolution increases. The response is more mixed for the intense storms although the biases are mostly not statistically significant."**

For maps of high wind/gust percentiles there are differences between the models, especially over areas with high elevation, but since we apply bias correction, the bias in the raw data will be different to that in the event set therefore we feel it may be confusing to include it in this paper. The issue is apparent from Figure B1 of the original manuscript, so we believe adding the extra figure will not add much benefit. It is, of course, an important and interesting issue, but we are working on another paper examining the differences in the North Atlantic storm track in these models from a more physical perspective, so information on the raw model wind speed will fit better in that study.

**RC1.4** The authors are using a loss index derived by Klawa and Ulrich (2003). Although this index is widely used, there could be other possible formulations for a loss function. For example: power law damage function with higher exponents than 3. For example:

Prahl, B. F., Rybski, D., Burghoff, O., and Kropp, J. P.: Comparison of storm damage functions and their performance, Nat. Hazards Earth Syst. Sci., 15, 769–788, https://doi.org/10.5194/nhess-15-769-2015, 2015.

Higher exponents could amplify LI differences between the used climate models and could have large impact in estimations of future losses, when wind speeds might get higher. I would not expect a detailed evaluation of possible effects, but a least a discussion.

This is a good point, which we will mention in Section 3.2.

"**We note that other loss indices exist (see Prahl et al., 2015, for example), and those with exponents greater than three may amplify differences between models and re-analysis compared to the results presented here.**"

**RC1.5** What is „similar strength"? For example: In the case of Daria the LI is 6x or 2x larger than the PRIMAVERA model footprints. I think it could be helpful to see a selection of model footprints, which are really close to the Daria´s LI. Furthermore, I would like to see footprints of events, that are close to the 200 year event. Do they look real?

We have updated Figure 5 of the original manuscript to include footprints which have a more similar LIs to Daria and Klaus, and to include the footprints with return periods of ~200, ~100 and ~50 years (since the data set is 1332 years, these are the 7th, 13th and 27th strongest storms). Although the footprints appear reasonable, there is large uncertainty on the most extreme gusts (as described in Section 5 of the paper). The new figure is shown below (Figure R5 in this document). We will add the following to the text (L) :

"**Also shown in Figure 5 are the footprints for the storms with approximate return periods of 200 yr, 100 yr and 50 yr. The footprint of the 200 yr event is truncated indicating it may be part of a complex cluster of storms. The 100 yr and 50 yr events are both large scale events over northern Europe, with footprints resembling that of Daria. There are small areas of very extreme gusts around Benelux, Germany and Poland, whose magnitude should be considered uncertain due to the bias correction (see Section 5).**"

**Referee 2**

**RC2.1** L112 – You have chosen to use the atmosphere-only simulations of the PRIMAVERA models. What is the justification of using these instead of fully coupled simulations?

We are working on another study comparing the North Atlantic storm track biases between the coupled and AMIP simulations, and the coupled models appear to have stronger biases, possibly due to an area of negative SST bias in the North Atlantic (see Athanasiadis et al., 2022, submitted), so we felt it would be beneficial to look at AMIP models first before adding the additional complexity and differences that arise from ocean variability.

**RC2.2** L124-125 – For the models that output gust data, how does your final diagnostic compare to the model output gusts? Are they at all similar?

Some of the models use a gust parametrisation which is dependent on the horizontal wind speed, therefore biases in the winds are also present in the gusts.  This is an advantage of the method we have chosen, as it implicitly bias corrects.  We expect there to be differences between the two methods over high land in particular.

**RC2.3** L289 – Why this threshold? Is it a particular threshold from the reanalysis LI or has it been arbitrarily chosen by the authors?

It was chosen because the named historical events given in the XWS catalogue (Roberts et al., 2014) have LIs of the order 1e6.  We will clarify this in the text at L289:

"(defined as those with LI > $1.0 \times 10^6$, **based on the LIs of the named historical events in Roberts et al. (2014); these storms** occur approximately once every two winters over Europe and make up 70% of total losses).  "

**RC2.4** L298 and Fig. 4 – How does the track density of the severe storms compare to that of all the storms? As the severe storms are what you are interested in would this be more appropriate? Are the biases of the same magnitude and in the same locations?

This is a good point.  We will include the track density biases for the severe storms (LI>1e6) in Figure 4.  The updated Figure 4 is shown in Figure R2 of this document.  We will mention in the text at L302:

*"The increase in storm numbers with resolution is also reflected in Fig. 4, which shows the track densities for footprints which have non-zero LI**, and for intense storms only**.  The maximum track density is located over the UK, which is expected given the area used to calculate the LI (Fig. 1), and the fact that maximum winds tend to occur south of the tracks.  The underestimation of non-zero LI tracks is most pronounced over the UK and the western parts of the European continent, but the bias is much reduced in the higher resolution models.  **For severe storms the bias in track density is mostly statistically insignificant over western Europe, but there is a slight over-estimation in storm numbers in the eastern Mediterranean basin."***

**RC2.5** Fig. 4 – are these biases significant?

We have now marked on Figure 4 where the differences are significantly different from zero with 95% confidence according to Welch's unequal variances t-test (see Fig R2 below). We will mention this in the captions for the figures.

**RC2.6** L304-305 and Fig. 5 – this sentence needs re-phrasing/clarifying. In the caption it is clear you are showing footprints of similar structure (as far as I can tell). However the sentence states that you are showing those of similar LI, which is not the case. On this note i think it would also be very useful to have a comparison of similar LI storms as Daria, Kyrill, Anatol,etc.

Please see reply to RC1.5 above. We have now replaced some of the footprints in Figure 5 to include ones with an LI more comparable to the historical storms (Figure R5 in this document).

**RC2.7** L305 – How did you select these storms? Was this just done by eye or was there a quantitative measure to select them?

We first selected footprints with LIs within a particular range (in the original manuscript we selected LIs > 1e6, but in the updated version we have narrowed the range to be closer to the values of the historical storms we are trying to match). Then we calculated the spatial correlation coefficient between the subset of footprints and the historical storm. Since the spatial correlation is a rather rough estimate of a pattern match, we then examined the footprints with the highest correlation coefficients by eye to select the best looking matches.

**RC2.8** Table 2 and onwards – How does the inclusion of the MM HadGEM3 run affect the model bias? Does this overweight the means toward the HadGEM3 climate. Please quantify the impact of having both the HM and MM simulations compared to just having HM.

We will now include the track densities for individual models in the appendix (for all storms with a non-zero LI, and intense storms only). Please see Figures R3 and R4 in this document. It can be seen that all models show a similar biases and change with resolution so we believe the multi-model means in Table 2 and Figure 4 are a fair representation of the data.

**RC2.9** L345-347 – these storms that are discussed in Sect. 5 it would be good to show all the evidence behind removing these from the data. LI footprints could be included in the appendix.

When plotting the LI against return period for individual storms it is clear that these three storms are outliers (see Figure R6). Further inspection of the footprints revealed clusters of grid points with very extreme gusts centred over large population areas. As discussed in Section 5 of the manuscript, the uncertainty on the bias correction of the most extreme gusts is very large and it seems likely that it has resulted in a large over-estimation of the gusts at these grid points.

We will clarify the justification the text at L347 as follows:

*"Note that three model storms (listed in Appendix C) had to be removed from the seasonal aggregate losses as they were considered unrealistically extreme. **They are clear outliers when plotting LI against empirical return period for individual storms,** and their inclusion prevented a satisfactory GPD fit. **The extreme LIs are due to clusters of grid points with extreme gusts occurring over large**

*population centres, and are a result of the bias correction method used (discussed further in Sect. 5)."*

**RC2.10** L352-354 – This statement assumes that the distribution of LI is the same in the reanalysis as the models, which it may not be - it may also be good to do the GPD fit to the reanalysis data in Fig. 7b to compare to the fit of the reanalysis and models.

We will modify the text to:

*"**Assuming the model LI distribution is representative of observations,** the GPD fit estimates that the most extreme season over Europe in re-analysis (1989/90), which had a total LI of 4.5⬚107, has a return period of 75-200 years under present day conditions, **somewhat** longer than the 35 years estimated from the re-analysis data alone."*

Since the GPD fit for the models is performed for seasons with aggregate losses >90th percentile, it is not possible to fit the re-analysis in the same way (there would only be 4 data points). This is why we chose to show the model data is consistent with re-analysis by sub-sampling 35 year time series from the model data, rather than attempting to fit the re-analysis data and estimating the uncertainty on that.

**RC2.11** L387-391 – This dispersion value is this for cyclone counts in your entire European region? In Mailier et al. (2006) this is shown as a spatial field. Do your results compare spatially to this?

That is correct. The spatial distribution of the dispersion is an interesting question, although it would not be directly comparable to Mailier et al, because our analysis is for intense storms as measured by the LI, which have a different spatial distribution to all tracked storms (or tracked storms subsetted by intensity measures such as minimum mean sea level pressure). We therefore feel it is not appropriate to include in this paper, but this could be a question for future study.

**RC2.12** Fig. 9 – are figs 9a and 9b consistent for other geographic regions as is the case in Fig. 9b?

Yes, the figures are consistent for the countries given on the axis of Figure 9b.

**RC2.13** What is the impact of the 5 different models in your analysis? Do they all exhibit different behaviour? Are the results shown for example in Figs. 6, 8, 9 an artefact of the model mean or is this evident as a feature of each of the individual models?

We have plotted Figures 6, 8 and 9 for individual models in Figures R7-R9 of this document. You can see that in general the same behaviour is evident in the individual models. Fig R7 shows the LI distribution for each model is consistent with observations. The dispersion parameter to measure the storm clustering shows quite large variation between each model (Figure R8), but this is likely because of the large uncertainty in this measure when the number of years, N, is small (N~64 for models with only one ensemble member). We have estimated the 95% confidence intervals for the dispersion using a bootstrapping method (randomly re-sampling the model data with replacement to create 1000 time series of length N winters, and calculating the dispersion for each re-sampled time series), and do indeed find large confidence intervals (given in panel titles in Figure R8). Fig R9

shows that all models generally reproduce the expected geographic relationship between aggregate seasonal losses and NAO, with positive correlations in northern European countries, and negative correlations in southern European countries.

We will mention the consistency of the results in individual models in the conclusions at L503:

"**The data presented in this paper is for the multi-model ensemble, but similar conclusions are reached when looking at individual models.**"

**RC2.14** You state the data is available from the author, however, I think this dataset would be very beneficial for general access. Are there plans to make this publicly available?

This is a good point.  We have now uploaded the data to Zenodo where it is publicly available (https://doi.org/10.5281/zenodo.6492182).  We will mention this in the abstract and conclusions.

**RC2.15** Some of the references quoted in the text are not included in the reference list – please double check the reference list.

Thank you for spotting this.  We will correct the reference list.

**Referee 3**

**RC3.1** L 40: Can you provide a reference for this definition?

Yes – we will add a reference to Haylock et al (2011; doi:10.5194/nhess-11-2847-2011) at L41. Apologies for this omission!

**RC3.2** L 58 ff: you are describing the use of dynamical models to generate event sets. Disadvantages are e.g. coarse resolution of the models with all its difficulties and pit falls (too zonal storm track, too small latent heat release, etc). But you are also mentioning the WISC event set and studies using ensemble prediction systems where the horizontal resolution is comparable to the used PRIMAVERA models. Can you explain the reason and advantages of using your model ensemble in comparison to those studies?!

One of the main benefits of the PRIMAVERA event set over WISC is that it is made from a multi-model ensemble.  We have also attempted to separate the storms in order to study clustering, and have applied a different bias correction method.  But overall we would say this event set is not intended to replace existing ones, but rather complement them, and will allow users to explore uncertainty arising from constructing event sets with different methods.

We will mention this in the text at L103:

*"The aim of this paper is to describe the method used to create the event set from PRIMAVERA models and to show how it compares to re-analysis.  The method involves first identifying the storms using a tracking algorithm, then extracting the model surface winds associated with each storm to make the footprint.  The footprints from different climate models are then re-gridded to a common 0.25°✕0.25° grid, and the model winds are bias corrected and converted to gusts using quantile mapping.*  **We note that the concept of this event set is similar to the stochastic event set created for the WISC project (Steptoe, 2017).  The main differences between the two event sets are (i) use of a multi-model ensemble for PRIMAVERA rather than a single model ensemble; (ii) separating the footprints of storms occurring in the same 72 hour period in order to study temporal clustering; and (iii) applying different bias correction methods.**  *The model and re-analysis data used are described in Sect. 2, and Sect. 3…"*

**RC3.3** L 124: see first comment. There is no reference for this definition.

See response to RC3.1.

**RC3.4** L 137: what does it mean if you are writing that track were unavailable at the time? Are the tracks provided by a computing center? Couldn't you perform the tracking by yourself?

The tracking was performed as part of another PRIMAVERA work package.  The tracking on ERA5 had not yet been completed when we needed to generate the re-analysis footprints for a project deliverable so we had to work with what was available.  As generating the footprints takes a considerable amount of time and resource and use of ERA5 tracks rather than ERA-Interim is expected to make such little difference (see L138), it was decided it would not be worth updating the re-analysis footprints at this stage, but it would be good to do this is further work is done.

**RC3.5** L142: you are mentioning the definition of the footprint again and arguing that you want to comply with industry standard. Reference would help

See response to RC3.1.

**RC3.6** L 143: what is the reason for exactly this definition of the domain?

It covers the region most affected by extra-tropical cyclones formed over the North Atlantic. (It is also the same region as used in the WISC project.)

**RC3.7** L 146: what does „central day of the 72h period" mean? Is it 36h before and 36h after this day? If this is the case, this information is very important. Please use this shortly, when mentioning the 72h period and the connection to a cyclone track the very first time. I was wondering before how to connect a 72h period to a cyclone which lives for a couple of days.

Since the footprints are made from daily maximum data, the 72 hour period has to cover three days from 00Z on day 1 to 00Z on day 4. The central day is day number 2 of that period. We have tried to clarify this in the text (L146):

*"One 72-hour **(3-day)** footprint is produced per track, despite typical track lengths being longer than 72 hours.  **Since daily data is used, each 72 hour period runs from 00 Z on day 1 to 00 Z on day 4.** Following Roberts et al. (2014), for each track, the central day **(day 2)** of the 72-hour period over which to take the maximum gusts is identified by finding the day of the maximum…."*

**RC3.8** L 185: I am wondering about the time frame you had to finalize your study. In how far the results are influenced or biased by this reduced set of 12 winters?

As there are 1332 winters remaining in the study, the reduction of 12 winters should make little difference.

**RC3.9** L 192ff (Fig 2): I do not understand how the mask is calculated you are using to separate footprints. Since you are writing to consider gridpoints less than 1500km away from the cyclone track, I expect an area (tube shape) around the track. That seems not to be the case. Can you explain why?

This is because there are often several extra tropical cyclones in the domain on a given day, so each grid point is assigned to the storm track point that it is nearest to (and if it is >1500km away from a track point it is not assigned to any storm). This is explained at L190 and illustrated in Figure 2.

**RC3.10** L 206: does this reference define footprints as 3-s gust over 72h? This would be important to use earlier as commented before.

That reference does not mention 72 hours specifically, but the Haylock (2011) reference mentioned earlier should cover this.

**RC3.11** L 209: At L 133 you are writing to use hourly maximum gusts of ERA5. Are those representative for 3-s gusts since you are using this as „observations"

The hourly maximum gusts are actually the maximum 3-s gust obtained over the hour. We apologise for the confusion. We will clarify this at L133 as follows

"Hourly maximum **3-s** gusts for October–March were extracted and converted to daily maxima…"

We have also added some discussion here as to how well ERA5 gusts compare with station observations (see response to RC1.2 above).

**RC3.12** L 268: What is the source for your population density?

It is from Gridded Population of the World, Version 4 (GPWv4) (Center for International Earth Science Information Network - CIESIN - Columbia University. 2016. Gridded Population of the World, Version 4 (GPWv4): Population Count. Palisades, NY: NASA Socioeconomic Data and Applications Center (SEDAC).)  We will add this reference to L272.

**RC3.13** L 292ff: The comparison is done on the same grid, isn't it? Model wind speed is statistically downscaled to the ERA5 grid. Can you explain the mechanism why the coarse resolution underestimates LI?  Do you have an explanation why the resolution effect is cannot be seen anymore for severe footprints?

It is correct that the comparison is done on the same grid.  It is interesting that despite the bias correction the lower resolution models still underestimate the number of storms.  This must be because the bias correction is applied to daily data at each grid point independently, but storms cover large areas.  In order to get the correct number of storms, the models must simulate their spatial distribution correctly.  A low resolution model may not be able to simulate small scale features in a storm where there are a small number of grid points exceeding the 98$^{th}$ percentile.  We have attempted to explain why this is at L294 of the original manuscript:

"*The number of footprints with a non-zero LI tends to increase with model resolution, possibly because to have LI>0 there must be regions with wind speeds greater than the local 98th percentile, which may occur in a higher proportion of storms if small scale features embedded with high wind 295 speeds are better resolved.*"

We are unsure why the effect is not seen so much for severe footprints.  It could be because these tend to be larger scale anyway, so resolving of small scale features is less important.

**RC3.14** L 335: it is hard to compare the distribution of ERA and PRIMAVERA especially for high LI values. Would it be beneficial to use CDF? Additionally it has the advantage to be independent of bin width.

We have plotted the cdf in Figure R10 but we prefer the distribution as shown in Fig 6 as it gives the additional information about the number of storms per season, and shows that the models can produce a realistic number of intense storms, which is important information.

**RC3.15** L 343: Return periods for ERA are not calculated with a GPD fit but empirically, isn't it? That means that the most intense season (which is 89/90) has return periods of the length of the time series, i.e. 35 years. Maybe it is worth to shortly explain this just to avoid misunderstanding.

That is correct.  We will clarify it in the text at L343:

"Figure 7(a) shows the **empirically estimated** return period**s** for seasonal *aggregate* losses (seasonal sum of LI) in the PRIMAVERA models and the re-analysis data."

**RC3.16** L 345: I do not understand the return periods of the unrealistically extreme events (open circles).

For the seasons which contained the unrealistically extreme storms we calculated two aggregate losses – one including the extreme, and one where it was removed. We have plotted both values at the empirical RP calculated after removing the extreme from the aggregate, so the open circles are directly above the data for the same season with the extremes removed. We did it this way because if we re-calculated the RPs for the aggregates including the extremes, all the data points get re-ordered and it is unclear to see the effect of including the extreme in the individual seasons. In other words, the plot as it is shows that when the extremes are removed from the aggregate losses for the three seasons their RPs fall to ~50 yr (compared to the 1332 yr, 666 yr, and 444 yr return periods they would have had otherwise, being the strongest three seasons in the dataset). We have tried to clarify this in the caption to Fig 7(a):

*"Figure 7: Return period curve for seasonal aggregate losses. The GPD fit to PRIMAVERA data is shown by the red line, with individual seasons shown by the red dots. The open red circles show the aggregate losses for the three seasons which contained the unrealistically extreme storms, before these storms were removed from the aggregate **(plotted at the same return period)**…."*

**RC3.17** L 455ff: there are different uncertainties for the use of the empirical method or the GPD for the bias adjustment. Is it possible to take this into account when calculating return periods in Fig 7? The confidence interval in Fig 7 is just due to the GPD fit of the PRIMAVERA LI data but the uncertainty to calculate the LI is not taken into account. At least it would be worth to discuss it.

This is indeed what we are trying to do! By showing the difference in LI as a function of RP we see that the uncertainty the GPD fit does not cover the uncertainty arising from different bias correction methods for large return periods because the blue points lie outside the red shading. We will clarify this in the text in the discussion at L470:

"**It also shows that the uncertainty in the GPD fit to the model data does not cover the uncertainty arising from the LI values themselves.**"

**Figures**

[Figure]

Figure R1: Plots of observed maximum 3-s gusts taken over a 72 hour period against ERA5 gusts over the same period linearly interpolated to the station location, for a selection of famous historical storms. The maps show the ERA5 maximum wind gusts over 72 hours for the whole field, with black dots representing the station locations. The dates used for the storms are Lothar: 25th-27th Dec 1999; Kyrill: 17th-19th Jan 2007; Anatol: 2nd-4th Dec 1999; Daria: 24th-26th Jan 1990; Klaus: 23rd-25th Jan 2009; Jeanette: 26th-28th Oct 2002. The blue line in the scatter plots is *y=x*. The bottom panels shows all the data points plotted on the same axes with the *y=x* line in black.

[Figure]

*Figure R2: Update to Figure 4 of original manuscript, now including track densities of intense storms (LI>1e6) in panels (g)-(i), and their bias ((k), (l)), and improvement from LR to HR (j). The yellow contour marks where the bias is statistically different from 0 with 95% confidence according to Welch's unequal variances t-test.*

[Figure]

*Figure R3: Track density bias (model – ERA) for storms with a non-zero loss index over Europe for individual models, for the period Oct-Mar 1979/80-2013/14. The yellow contour marks where the bias is statistically different from 0 with 95% confidence according to Welch's unequal variances t-test.*

[Figure]

*Figure R4: As for Figure R3, but intense storms only (LI>1e6).*

[Figure]

*Figure R5: Updated version of Figure 5 in the original manuscript. Model footprints have been updated to include ones with more similar LIs to the historical storms, and the bottom row ((m) to (o)) shows the footprints for events with RPs of approximately 200 yr, 100 yr and 50 yr (since the data set is 1332 years, these are the 7th, 13th and 27th strongest storms).*

[Figure]

*Figure R6: Plot of loss index against empirically estimated return period for individual storms in the PRIMAVERA event set. The stars mark the LI of the three storms which were removed from the aggregate losses in Figure 7 of the original manuscript.*

[Figure]

*Figure R7: Distribution of number of severe storms (LI>1×10⁶) per extended winter as a function of LI in individual PRIMAVERA models (red) and re-analysis (black). Vertical red lines show the 95% range in frequency estimated from 1000 35 year samples (with replacement) from the model data. Note that the last LI bin (LI>17×10⁶) is larger.*

[Figure]

*Figure R8: Distribution of number of severe storms per winter in individual PRIMAVERA models (red) and re-analysis (black). The red lines on each bar show the 95% range of season counts for 1000 35 re-sampled years of PRIMAVERA data. The dispersion parameter estimated from each model is given in the panel titles, with 95% confidence intervals estimated from 1000 random re-samples of the data (with replacement).*

[Figure]

*Figure R9: Rank correlation coefficients between seasonal aggregate LI and NAO over the countries in the European domain for individual PRIMAVERA models (red dots) and re-analysis (black dots). The vertical red solid lines indicate the 95% distribution of correlations from 1000 35 year samples from PRIMAVERA data (not the confidence intervals on the correlation coefficient of all 1332 years of data), to show consistency with re-analysis.*

[Figure]

*Figure R10: CDF of model (red) and observed (black) storms as a function of LI, for storms with LI>1e6.*

---

## Author Response (AR2)

Dear Editor,

Thank you to you and the referees for taking the time to re-review our manuscript.  We have made the technical corrections as requested:

*Table 1 - The nominal resolution quoted for CNRM-CM6-1-HR atmosphere is 100km, and not 50km as quoted in the text (https://wcrp-cmip.github.io/CMIP6_CVs/docs/CMIP6_source_id.html)* This has been updated to 100 km.

*L137 – bold text* Changed to normal font.

*L303 - struck through text* Removed.

*L280 - I assume you use the 98th percentile of each model in its calculations - please clarify this at this point in* That is correct.  We have added to L280 "(calculated separately for each model)".

In addition to these changes, we have updated Fig 3 and Fig B1 to be more readable by readers with colour vision deficiencies, as was advised during the file upload validation.

Best regards,

Julia Lockwood and co-authors.